# Enabling Conversational Behavior Reasoning Capabilities in Full-Duplex Speech

## Abstract

Human conversation is organized by an implicit chain of thoughts that manifests as timed speech acts. Capturing this causal pathway is key to building natural full-duplex interactive systems. We introduce a framework that enables reasoning over conversational behaviors by modeling this process as causal inference within a Graph-of-Thoughts (GoT). Our approach formalizes the intent-to-action pathway with a hierarchical labeling scheme, predicting high-level communicative intents and low-level speech acts to learn their causal and temporal dependencies. To train this system, we develop a hybrid corpus that pairs controllable, event-rich simulations with human-annotated rationales and real conversational speech. The GoT framework structures streaming predictions as an evolving graph, enabling a multimodal transformer to forecast the next speech act, generate concise justifications for its decisions, and dynamically refine its reasoning. Experiments on both synthetic and real duplex dialogues show that the framework delivers robust behavior detection, produces interpretable reasoning chains, and establishes a foundation for benchmarking conversational reasoning in full duplex spoken dialogue systems. Project page: `https://got-duplex.github.io/`

## 1 Introduction

Recent advances in spoken dialogue systems have shifted from turn-based, half-duplex models to full-duplex systems capable of simultaneous listening and speaking (Arora et al., 2025b; Nguyen et al., 2022b; Inoue et al., 2025). As illustrated in Figure 1 (left), the dominant paradigms frame this task as prediction. The first approach, Next Segment Prediction, models the agent's response as a complete turn (Hara et al., 2018; Li et al., 2022; Lee & Narayanan, 2010). A more recent approach, Next Dual-Token Prediction, generates simultaneous token streams for both speakers to better handle overlap and real-time interaction (Nguyen et al., 2022a; Défossez et al., 2024). While these methods have improved system responsiveness, they treat conversation as a sequence generation problem, bypassing the cognitive layer of reasoning that governs human interaction.

Human conversation, however, operates on a more abstract and causal level. We argue for a paradigm shift from black-box prediction to an explicit process of Next Behavior Perception and Prediction, as depicted in Figure 1 (right). When Speaker 1 produces an utterance, Speaker 2 does not simply predict the next sequence of words. Instead, they first perceive the behavior (e.g., recognizing a constative speech act), which triggers an internal chain of thought (e.g., deciding not to interrupt and to remain silent). This reasoning process culminates in a generated action (e.g., an acknowledgement). This gap between pattern matching and causal reasoning is a fundamental barrier to creating truly natural AI agents. Our work addresses the core scientific question: How can a machine model this perception-reasoning-generation loop to make principled, interpretable decisions in real time?

To tackle this challenge, we introduce a framework that operationalizes the process. Our approach is twofold. First, we formalize the Perception stage with a hierarchical conversational behavior detection model. This module learns to identify conversational behaviors at two dimensions: high-level speech acts (e.g., *constative, directive*) (Jurafsky & Martin, 2025) that capture communicative intent, and low-level acts (e.g., *turn-taking, backchannel*) that describe interaction mechanics (Schegloff, 1982; Gravano & Hirschberg, 2011; Duncan, 1972; Raux & Eskenazi, 2012; Khouzaimi et al., 2016; Marge et al., 2022; Lin et al., 2025; Arora et al., 2025b; Nguyen et al., 2022b). This provides the system with a structured understanding of the ongoing dialogue. Second, we model the explicit rea-

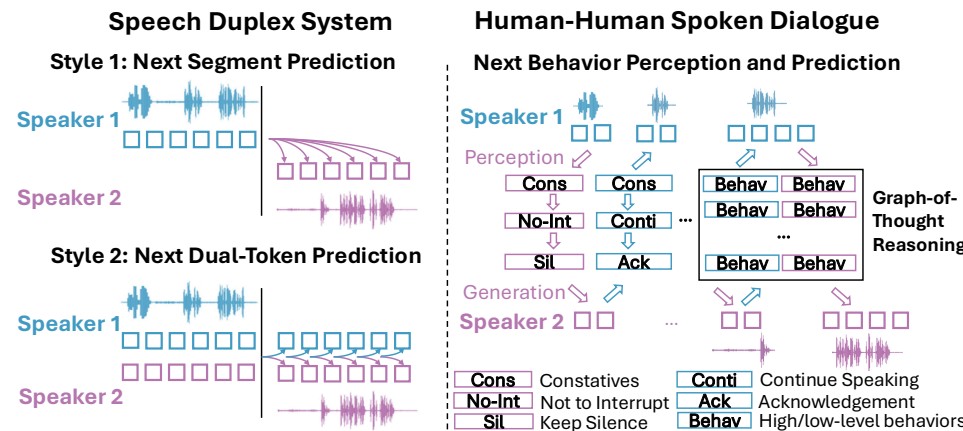

Figure 1: **Comparison of dialogue paradigms.** (Left) Traditional duplex systems frame conversation as a direct sequence prediction task. (Right) We propose a framework based on next-behavior perception and reasoning, where an agent perceives the speaker's act, reasons using a Graph-of-Thoughts, and then generates a response.

soning process with a Graph-of-Thoughts (GoT) system (Yao et al., 2024). This system constructs a dynamic causal graph from the sequence of perceived speech acts, capturing the evolving chain of thought within the conversation. By performing inference over this graph, our model can not only predict the most appropriate subsequent behavior but also generate a natural language rationale explaining its decision. This transforms the opaque prediction task into an auditable reasoning process, which provides a unified benchmark for *evaluating conversational behavior in duplex speech systems.*

To train our framework, we developed a hybrid corpus that combines behavior-rich simulated dialogues with real-world conversational data annotated with human rationales. Our analysis confirms that the synthetic data reproduces key interactional structures of human conversation, such as turn-taking dynamics.

In summary, our contributions are:

- A conceptual reframing of full-duplex interaction from next-token prediction to next-behavior reasoning, arguing that modeling the causal chain from intent to action is critical for natural dialogue.
- A hierarchical speech act detection model that perceives conversational behaviors at both high (intent) and low (action) levels, serving as a foundational module for reasoning-driven dialogue systems.
- A GoT framework for conversational reasoning that models intent-act dynamics as a causal graph, enabling real-time, interpretable decision-making and rationale generation.
- A comprehensive empirical validation demonstrating that our system effectively detects conversational behaviors, generates plausible rationales, and successfully transfers its reasoning capabilities from simulated to real-world full-duplex audio.

## 2 RELATED WORK

**Duplex Models.** Recent work in spoken dialogue systems (SDMs) increasingly draws on human conversational behaviors. Building on insights from human conversation, recent SDMs have progressed toward duplex capabilities—systems that listen and speak concurrently. SDMs are commonly built in two modes. Half-duplex models follow a turn-by-turn protocol, waiting for explicit end-of-turn signals (e.g., end-pointing/VAD) before responding, which simplifies streaming but adds latency and suppresses natural overlap. They commonly adopt next segment prediction (Hara et al., 2018; Li et al., 2022; Lee & Narayanan, 2010) where the system predicts the agent's response as a segment. Full-duplex models listen and speak simultaneously, modeling overlapping speech, micro-pauses, and background noise to sustain context and deliver timely cues (e.g., continuers, barge-in handling). They either employ the next segment prediction paradigm (Arora et al., 2025b; Nguyen

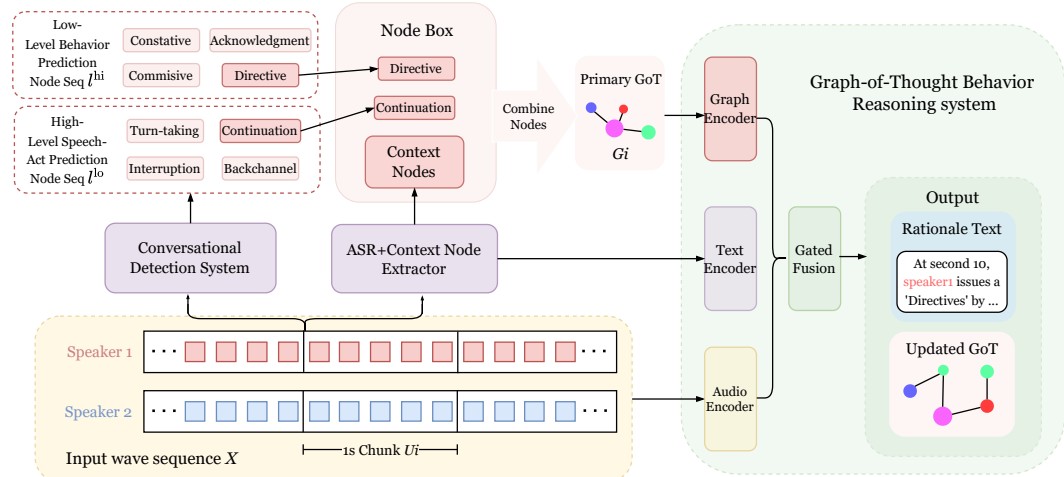

Figure 2: **Conversational Behavior Detection and Reasoning Framework.** When an audio clip is fed into the Conversational Detection System, the model segments the entire clip into 1-second chunks. For example, for the chunk shown in the figure, the Conversational Detection System labels the high-level behavior as *Directive* and the low-level speech act as *Continuation*. These two nodes, together with the context nodes extracted via OpenIE, constitute the primary GoT. In the Graph-of-Thought (GoT) Behavior-Reasoning System, the primary GoT, the transcript, and the raw audio are processed by separate encoders and then combined via gated fusion, producing the rationale text for this chunk as well as an updated GoT graph.

et al., 2022b; Inoue et al., 2025) or introduce the next dual token prediction (Nguyen et al., 2022a; Défossez et al., 2024) scheme to incorporate the listener's branch.

**Conversational Behaviors.** Interactive behaviors in SDMs have been extensively examined to foster mutual understanding, user engagement, and social connection between human and computer. However, recent studies in SDMs only focus on low-level behaviors (backchannel (Schegloff, 1982; Lin et al., 2025), turn-taking (Gravano & Hirschberg, 2011; Duncan, 1972; Raux & Eskenazi, 2012), interruption (Khouzaimi et al., 2016; Marge et al., 2022), and continuation (Lin et al., 2025; Arora et al., 2025b; Nguyen et al., 2022b)), which overlook the importance of discourse-level intent that drives these phenomena. To resolve this, we also introduce high-level speech acts (constative, directive, commissive, acknowledgment) (Jurafsky & Martin, 2025) to restore this layer, enabling more interpretable modeling and evaluation for conversational behaviors.

**Graph of Thought Reasoning.** Reasoning techniques such as Chain-of-Thought (Wei et al., 2022), Tree of Thought (Yao et al., 2023), and Graph of Thought (Besta et al., 2024) have help guide LLMs through intermediate steps to boost reasoning accuracy. Particularly, Graph of Thought enables arbitrary reasoning dependencies by modeling an LLM thought as a vertex, and dependencies among thoughts as edges, offering increased reasoning capabilities while reducing inference cost (Besta et al., 2024; 2025). Yao et al. (2024) adopts a two-stage framework: the first stage contains a GoT encoder for thought graph representation and a gated fusion to combine multimodal inputs and generate rationales, and the second stage generates text answers based on the rationale. GoT has also been used to solve math problems (Bai et al., 2025), real-world external graph question answering (Jin et al., 2024), and fact-retrieval and reasoning (Fang et al., 2024). This work is the first to use GoT to infer conversational behaviors in speech duplex systems.

## 3 CONVERSATIONAL BEHAVIORS DETECTION SYSTEM

Human-to-human spoken conversations involve both the perception and prediction of conversational behaviors. A key prior is that *humans rely on the perception of historical conversational behaviors*, which serves as the foundation of our duplex conversational behavior reasoning pipeline. Moreover, conversational behaviors unfold at multiple hierarchical levels: high-level acts such as constatives, directives, commissives, and acknowledgments (Jurafsky & Martin, 2025), and low-level acts such

as turn-taking, backchannels, interruptions, and pauses. Motivated by this structure, we propose a hierarchical conversational behavior detection system.

### 3.1 SYSTEM DESCRIPTION

As shown in Fig. 2, two-channel speech input $X = \{x_t\}_{t=1}^T \in \mathbb{R}^{2 \times T}$ is processed to predict hierarchical labels at the segment level in a streaming manner. The speech signal is downsampled to 16 kHz and segmented into 1-second chunks $U = \{U_i\}_{i=1}^N$, where $U_i = x_{B_{i-1}+1:B_i}$ and $B_i = i \cdot N_{\text{block}}$. We extract acoustic features $h_i^B \in \mathbb{R}^{768}$ using HuBERT (Hsu et al., 2021) and semantic features $h_i^E \in \mathbb{R}^{768}$ from Whisper transcripts (Radford et al., 2023). The features are fused via gating and processed by a causal Transformer to produce contextual representations $z_i$, which feed into parallel classification heads for high-level and low-level speech acts. Details can be checked at Appendix A.3.

### 3.2 TRAINING AND INFERENCE

We maximize the likelihood of hierarchical label sequences $L = \{(l_i^{\text{hi}}, l_i^{\text{lo}})\}_{i=1}^N$ given the input:

$$P(L \mid X) = \prod_{i=1}^N P\left(l_i^{\text{hi}}, l_i^{\text{lo}} \mid U_{1:i-1}\right). \tag{1}$$

To address class imbalance, we apply inverse-frequency weights (Cui et al., 2019) computed on the training split:

$$\mathcal{L} = \sum_{i=1}^N \left[ \alpha \sum_j w_j^{\text{hi}} \cdot \mathbf{1}_{y_i^{\text{hi}}=j} \log p_{i,j}^{\text{hi}} + \beta \sum_k w_k^{\text{lo}} \cdot \mathbf{1}_{y_i^{\text{lo}}=k} \log p_{i,k}^{\text{lo}} \right], \tag{2}$$

where $\alpha$ and $\beta$ balance the contribution of high-level and low-level predictions, and $p_{i,j}^{\text{hi}}$ and $p_{i,k}^{\text{lo}}$ are the predicted probabilities for classes $j$ and $k$ respectively. During inference, we use a conditional independence approximation and condition on a fixed-length causal window $\hat{U}_i$ for computational efficiency:

$$P(L \mid X) \approx \prod_{i=1}^N P\left(l_i^{\text{hi}} \mid \hat{U}_i\right) \cdot P\left(l_i^{\text{lo}} \mid \hat{U}_i\right). \tag{3}$$

## 4 GRAPH-OF-THOUGHTS BEHAVIOR REASONING SYSTEM

To explain conversational behavior rationales, we construct a Graph-of-Thoughts (GoT) (Besta et al., 2024) that performs causal inference over predicted speech acts. Each behavior is represented as a node in a directed graph with edges denoting causal relationships, allowing the system to score behavior chains and provide interpretable rationales.

### 4.1 PROBLEM FORMULATION

Given a causal window $\hat{U}_i$ of length $W$ seconds ending at time $i$, let $S_i$ denote the incremental ASR transcript and $(\ell_i^h, \ell_i^l)$ the predicted high/low-level speech–act labels. We extract OpenIE triples $\mathcal{T}_i = \{(s_k, r_k, o_k)\}_{k=1}^{m_i}$ from $S_i$ and build a directed, integer–weighted graph

$$G_i = (V_i, A_i), \qquad V_i = \text{uniq}\Big(\{s_k, o_k\}_{k=1}^{m_i} \cup \{\ell_i^h, \ell_i^l\}\Big), \quad n_i = |V_i|.$$

Let $\{e_v \in \mathbb{R}^{n_i} : v \in V_i\}$ be the standard basis indexed by $V_i$. The adjacency counts subject$\rightarrow$object co-occurrences and adds self–loops:

$$A_i = I_{n_i} + \sum_{(s_k, r_k, o_k) \in \mathcal{T}_i} e_{s_k} e_{o_k}^\top \in \mathbb{N}_0^{n_i \times n_i}, \qquad (A_i)_{uv} = \#\{k : s_k = u, o_k = v\} + \delta_{uv}.$$

Each node $v \in V_i$ has a type $\tau(v) \in \{\text{text}, \text{sa-h}, \text{sa-l}\}$ and a $d$-dimensional embedding provided by the corresponding encoder:

$$\phi(v) = \begin{cases} E_{\text{text}}(v), & \tau(v) = \text{text}, \\ E_{\text{sa}}(v), & \tau(v) \in \{\text{sa-h}, \text{sa-l}\}, \end{cases} \qquad F_i = \left[\phi(v_1); \ldots; \phi(v_{n_i})\right] \in \mathbb{R}^{n_i \times d}.$$

## 4.2 System Description

Given audio input $X = \{x_t\}_{t=1}^T$, we process 1-second segments using a causal window $\hat{U}_i$ of size $W = 30s$. For each timestep $i$, we extract text nodes $\mathcal{V}_i^{\text{text}}$ (subject-relation-object triples from incremental ASR using OpenIE) and speech-act nodes $\mathcal{V}_i^{\text{sa}}$ (predicted high/low-level labels $(\ell_i^{\text{hi}}, \ell_i^{\text{lo}})$). The complete graph $\mathcal{G}_i = (\mathcal{V}_i, \mathbf{A}_i)$ combines both node types with adjacency matrix $\mathbf{A}_i$ encoding co-occurrence relationships. We encode the multimodal context using frozen encoders: HuBERT (Hsu et al., 2021) for audio ($h_i^{\text{a}} \in \mathbb{R}^d$), T5 (Raffel et al., 2020) for text ($h_i^{\text{t}} \in \mathbb{R}^d$), and GAT (Chen & Yang, 2021) for graphs ($h_i^{\text{g}} \in \mathbb{R}^d$). Features are fused via gating and processed by a causal Transformer to produce contextual representations $z_i$. More details can be checked in Appendix A.4.

## 4.3 Training Objective

The model learns to produce natural–language rationales $r_i$ that explain the predicted behavior at each timestep $i$ given the causal window $\hat{U}_i$ and the per–second graph $\mathcal{G}_i$. We maximize the conditional likelihood of the rationale set $R = \{r_i\}_{i=1}^N$ given the input $X$ by factorizing over timesteps and a T5 (Raffel et al., 2023) decoder generates each rationale sequence $r_i = (y_{i,1}, \ldots, y_{i,T_i})$ autoregressively from the fused representation $z_i$:

$$P(R \mid X) = \prod_{i=1}^N P\left(r_i \mid \hat{U}_i, \mathcal{G}_i\right), \qquad P\left(r_i \mid \hat{U}_i, \mathcal{G}_i\right) = \prod_{t=1}^{T_i} P_\theta(y_{i,t} \mid y_{i,<t}, z_i)$$

We train with teacher forcing by minimizing the token–level negative log–likelihood

$$\mathcal{L}(\theta) = -\sum_{i=1}^N \sum_{t=1}^{T_i} \log P_\theta\left(y_{i,t}^\star \mid y_{i,<t}^\star, z_i\right),$$

where $y_{i,t}^\star$ denotes the gold token and $y_{i,<t}^\star$ its prefix. Encoders for audio/text/graph are frozen; gradients update the fusion module that produces $z_i$ and the T5 decoder parameters $\theta$. We use standard subword tokenization, an EOS token to terminate $r_i$, and early stopping on validation NLL.

## 5 Experiments

### 5.1 Dataset

Both synthetic and real conversational speech are used to train and evaluate the behavior detection and GoT-based reasoning model, with train/validation/test sets partitioned into in an 8:1:1 ratio.

**Synthetic Dataset.** We generate dialogues from narrative prompts in ExploreToM (Sclar et al., 2024a) using GPT-4o (Hurst et al., 2024). The same model marks candidate backchannel and interruption points for event labels. Speech waveforms are synthesized with CosyVoice2 (Du et al., 2024), conditioned on timbre reference clips from LibriSpeech (Panayotov et al., 2015) to ensure voice consistency. For backchannel and interruption cases, we deliberately introduce controlled overlap between speakers. The resulting corpus comprises 28,000 clips totaling 192 hours. Event distribution shows turn-taking (11.4%), interruption (17.6%), backchannel (6.4%), and continuation (64.6%). In contrast to the Talking-Turns benchmark (Arora et al., 2025b), which reports lower proportions ($\approx 0.4\%$ each) for interruptions and backchannels in natural dialogue, we deliberately increase these events to provide stronger supervision signals. For Graph-of-Thoughts supervision, we generate per-second rationales using the OpenAI API (Achiam et al., 2023): for each second $t$, the model sees only text up to time $t$ and produces a rationale for that slice. All generated rationales were manually validated and corrected, yielding 37,100 rationale entries.

**Data Quality Check.** We analyzed turn-taking statistics of our simulation corpus and compared them against human reference dialogues and model baselines (dGSLM (Nguyen et al., 2022b) and Moshi (Défossez et al., 2024)). As shown in Table 1, our simulation data exhibit denser micro-segmentation than human dialogues, with higher IPU and pause counts per minute but comparable gap and overlap rates. Cumulative durations further suggest shorter overlaps and more within-speaker pauses, indicating frequent short backchannels or clause-internal hesitations rather than

Table 1: Turn-taking event frequencies (per minute) and cumulative durations (%) for the simulation dataset, a human reference, and model baselines.Human, dGSLM, and Moshi values are reproduced from Fig. 2 of (Arora et al., 2025b).

| Event type | *Number of events per minute* | | | | *Cumulative duration (% of time)* | | | |
|---|---|---|---|---|---|---|---|---|
| | **Simulation** | **Human** | **dGSLM** | **Moshi** | **Simulation** | **Human** | **dGSLM** | **Moshi** |
| IPU | 23.06 | 15.7 | 24.2 | 21.6 | 84.7 | 97.3 | 99.0 | 81.0 |
| Pause | 10.7 | 3.8 | 5.4 | 10.2 | 9.6 | 5.7 | 6.0 | 10.3 |
| Gap | 7.3 | 5.5 | 7.2 | 6.7 | 1.6 | 3.7 | 4.8 | 11.8 |
| Overlap | 6.7 | 6.6 | 10.9 | 4.8 | 4.2 | 6.7 | 9.7 | 3.1 |

Table 2: Speaking style metrics for the simulation dataset and a dGSLM baseline.

| Method | WPM | FWR |
|---|---|---|
| dGSLM (DLM-5) (Nguyen et al., 2022b) | 211.98 | 5.5 |
| Simulation Data | 240.8 | 6.89 |

long monologic turns. This pattern not only aligns with established observations of conversational floor management (Sacks et al., 1974; Stivers et al., 2009), but also indicates that our simulated data closely approximate the interactional structure of genuine full-duplex conversations. Additional quality measures, including speaking rate, noise level, and naturalness, are reported in Appendix A.9.1.

We also analyzed the speaking style of our simulation corpus. We align with dGSLM's speaking-style descriptors by reporting Words-Per-Minute (WPM) and Filler-Word Rate (FWR; per 100 tokens). Our simulation corpus shows WPM = 240.8 and FWR = 6.89. For reference, the strongest dGSLM variant (DLM-5) reports WPM = 211.98 and FWR = 5.5 under the same measurement recipe. The higher WPM indicates faster delivery than dGSLM generations, which can make interactions feel more responsive but also risks compressing IPUs (consistent with our elevated IPU/min). Meanwhile, a higher FWR suggests richer backchannel/hedge behavior ("uh-huh", "yeah", "okay"), which typically supports floor-keeping without taking the floor—again consistent with shorter overlaps and fewer/shorter gaps in our corpus statistics. Together, these style cues show that the simulation corpus is aligned with the conversational analysis tradition in which fillers and micro-overlaps serve as continuous grounding signals rather than overt turn grabs.

**Real Dataset.** We evaluate on real conversational speech from the Candor corpus (Reece et al., 2023). We curate a subset of 118 hours with existing timestamp annotations for backchannels, turn-taking, and continuation. Manual rationale annotations are added to assess model adaptability to real conversational data. In Candor, we implement the same process of annotating rationale text in the simulation dataset.

## 5.2 EXPERIMENTAL SETUP

We train the behavior prediction model and the GoT reasoning model separately on the same dataset but different annotations; the GoT model additionally leverages human-annotated rationales.

**Prediction Model Training.** We freeze all encoders and jointly train high- and low-level classifiers with missing annotations masked. For CANDOR, the low-level head uses three classes (no interruption labels). Class imbalance is handled with inverse-frequency loss reweighting.

**GoT Model Training.** We enforce strict causality using past-only windows ($W = 30$s). Whisper transcripts and OpenIE-based node extraction are completed during preprocessing to avoid runtime latency. Human-annotated rationales provide supervision signals during training.

Both models use AdamW optimizer with learning rate $3 \times 10^{-4}$, weight decay $10^{-2}$, linear decay with 1,000 warmup steps, and gradient clipping at 1.0. Training uses batch size 8 per GPU for 10 epochs with automatic mixed precision (fp16). Audio is resampled to 16 kHz and segmented into 1-second frames. Training takes approximately 48 hours for speech-act prediction and 7 hours for GoT generation on a single A6000 GPU. Results use random seed 42, with mean ± std computed over five independent runs for statistical analysis. More details can be checked at Appendix A.5.

Table 4: Performance on Synthetic Dataset: Per-class and Overall Metrics, which aggregate performance across all classes within each hierarchy level.

| High-level Speech Acts | | | Overall (High) | | Low-level Speech Acts | | | Overall (Low) | |
|---|---|---|---|---|---|---|---|---|---|
| Class | F1 (↑) | AUC (↑) | Metric | Score (↑) | Class | F1 (↑) | AUC (↑) | Metric | Score (↑) |
| Constatives | 0.705 | 0.833 | Macro F1 | 0.546 | Turn-taking | 0.710 | 0.953 | Macro F1 | 0.660 |
| Directives | 0.471 | 0.781 | Micro F1 | 0.585 | Interruption | 0.515 | 0.857 | Micro F1 | 0.768 |
| Commissives | 0.474 | 0.832 | Weighted F1 | 0.594 | Backchannel | 0.536 | 0.914 | Weighted F1 | 0.776 |
| Acknowledgments | 0.533 | 0.844 | Macro AUC | 0.822 | Continuation | 0.878 | 0.937 | Macro AUC | 0.915 |

## 5.3 Evaluation Metrics

We report classification accuracy for the speech act prediction and the generation accuracy for the GoT. We also report turn-taking event statistics for both the real and the synthetic datasets.

**Classification Accuracy.** Following prior work on turn-level event prediction, we evaluate speech act prediction with F1 (Manning et al., 2008) and ROC–AUC (Fawcett, 2006). The task is single-label multiclass; for each class, we use a one-vs-rest (OvR) scheme (Rifkin & Klautau, 2004). We report per-class F1 and per-class AUC in Table 4. We also report aggregated averages: macro-F1 (Sebastiani, 2002), weighted-F1 (Sokolova & Lapalme, 2009), micro-F1 (Sebastiani, 2002), and macro-AUC-OvR (Hand & Till, 2001) in Table 4.

**Generation Accuracy.** We evaluate generated rationales in the GoT model against human references on the validation set using BLEU-1 (Papineni et al., 2002), ROUGE-1/ROUGE-L (Lin, 2004), and the semantic SIMILARITY score (cosine similarity), shown in Table 5. For Moshi- and GPT-4–generated speech (no references), we rely on human ratings. We report 95% confidence intervals over multiple seeds for statistical reliability.

**Event Statistics.** We evaluate our simulation corpus with the standard corpus-level turn-taking statistics used by prior work: counts per minute of Inter-Pausal Units (IPU), Pauses, Gaps, and Overlaps, and the cumulated duration (percentage of the dialogue) for each event. All quantities are computed from VAD activity on the two channels, following the Talking-Turns protocol (Arora et al., 2025a). We adopt the dGSLM conventions (Nguyen et al., 2023). We report our dataset alongside the human reference used by Talking-Turns and dGSLM baselines and concrete values (including reproduced comparators) in Appendix A.9.1.

## 5.4 Baselines

For corpus-level baselines, we follow the *Talking-Turns* protocol and dGSLM conventions to compute corpus-level turn-taking statistics (IPU, Pause, Gap, Overlap). We compare our simulation data against Human performance, dGSLM, and Moshi. For conversational detection baselines, we compared the performance of our Conversational Behavior Detec-

Table 3: Detection Results on CANDOR.

| Class | AUC (↑) |
|---|---|
| Turn-taking | 0.796 |
| Backchannel | 0.701 |
| Continuation | 0.720 |

tion System on the Candor dataset and our simulation dataset. For its transferability we manually score its inference results of GPT4 and Moshi. For GoT behavior reasoning baselines, under a strictly-causal streaming regime with window $W \in \{10, 20, 30, 40\}$ s and look-ahead $L \in \{0, 5, 10\}$ seconds, we benchmark three modality configurations: **A** (Audio-only), **A+T** (Audio+Text), and **A+T+GAT** (Audio+Text+Graph). We keep $L = 0$ as a strict-causal baseline and include small $L$ as a latency-controlled reference.

## 6 Results

### 6.1 Conversational Behavior Detection Results

Detection results on synthetic data are presented in Table 4. Our model demonstrates strong ranking performance (AUC 0.78-0.95) but exhibits more variable classification accuracy (F1 0.47-0.88). Performance varies substantially across categories: constatives (F1: 0.705) and continuation (F1:

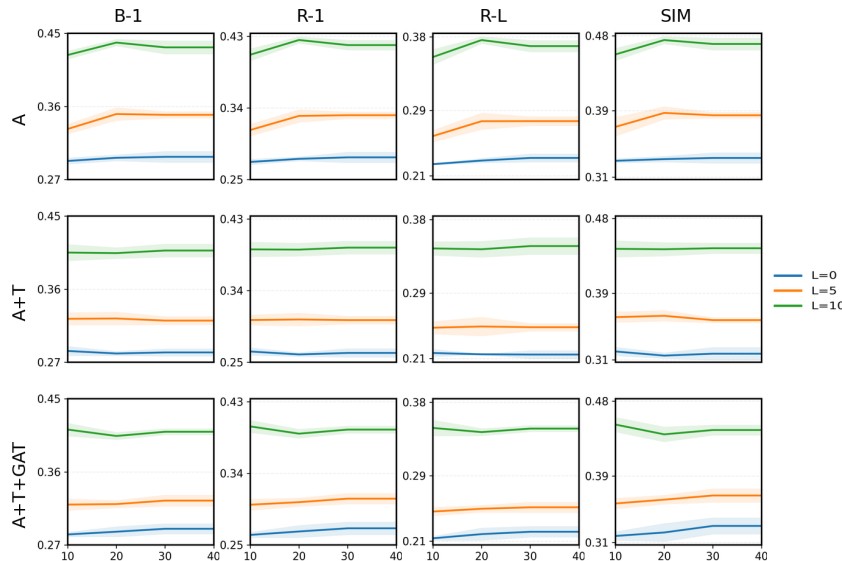

Figure 3: Window size $W \times$ look-ahead $L$ ablations under strict-causal streaming. Columns vary modality (A = Audio; A+T = Audio+Text; A+T+GAT = Audio+Text+Graph), and rows report BLEU-1, ROUGE-1, ROUGE-L, and cosine similarity (SIM). Curves show mean scores and shaded bands indicate 95% confidence intervals across seeds.

0.878) achieve the highest performance, while directives (F1: 0.471), commissives (F1: 0.474), interruption (F1: 0.515), and backchannel (F1: 0.536) show considerably lower scores. This pattern suggests that certain speech acts may be inherently more challenging to classify than others, potentially due to imbalanced data generation processes. Results on real speech data from CANDOR are shown in Table 3, where consistently high AUC scores are observed across categories. These results establish a foundation for the subsequent GoT-based graph reasoning stage.

## 6.2 CONVERSATIONAL BEHAVIOR REASONING RESULTS

We apply GoT to both simulated speech and real CANDOR data and evaluate the generated rationales, with results summarized in Table 5. The GoT pipeline achieves medium-to-high readability across datasets: BLEU-1 and ROUGE-1/L scores fall within the 0.42–0.58 range, while semantic similarity improves from 0.52 on synthetic data to 0.66 on CANDOR. All four metrics show improvements

Table 5: GoT pipeline results.

|  | Synthetic | CANDOR |
|---|---|---|
| BLEU-1 ($\uparrow$) | 0.480 | 0.580 |
| ROUGE-1 ($\uparrow$) | 0.470 | 0.560 |
| ROUGE-L ($\uparrow$) | 0.420 | 0.490 |
| Similarity ($\uparrow$) | 0.520 | 0.660 |

on CANDOR, with absolute gains of +0.07–0.14 (17–27% relative), the largest observed in semantic similarity. We attribute these improvements to *more natural conversational structures and discourse cues in real dialogues, which enhance the reliability of reasoning chains captured by GoT*.

**Ablations: Steaming and Multimodal Setup**   As joint behavior detection and reasoning operating in a streaming manner, we conduct ablations by varying the sliding window $W \in \{10, 20, 30, 40\}$ s, the look-ahead $L \in \{0, 5, 10\}$ s, and the input modalities (audio, text, graph), and report GoT scores in Figure 3. Results show that (i) small look-ahead ($L = 5$–$10$ s) stabilizes tokenization and improves accuracy without violating causality, (ii) medium windows ($W = 20$–$30$ s) best trade off recency and context, and (iii) naïve text addition can hurt performance, while incorporating the GoT graph with conservative gating recovers stability at low latency. Details can be checked at A.7.

## 6.3 CONVERSATIONAL BEHAVIOR REASONING ON FULL DUPLEX MODELS

We conduct a human evaluation of GoT's prediction rationales on full-duplex conversational audio, comparing our simulation corpus, GPT-4, and Moshi. Human raters judged whether rationales (i)

identified the correct speech act, (ii) inferred a plausible intent, and (iii) maintained discourse-level coherence, with scores given on a 1–10 scale. Results in Table. 6 show simulation achieves the highest mean score (8.93) > GPT-4 (7.06) > Moshi (4.25). Simulation rationales, though more abstract, consistently hit the correct acts and stabilize early. GPT-4 produces more concrete, locally grounded rationales but with slightly higher error, while Moshi underperforms due to omission of key local cues and greater temporal volatility. Overall, simulation provides stable abstraction, GPT-4 balances concreteness with robustness, and Moshi exhibits coherence drift. Overall, *GoT rationales trained on simulated dialogues transfer effectively to real audio*. Details can be checked at Appendix. A.8.

# 7 CONCLUSION AND LIMITATIONS

In this paper, we explored an approach to full-duplex conversation centered on causal reasoning rather than direct sequence prediction. We presented a framework that first perceives conversational behaviors at both an intentional and mechanical level, and then uses a Graph-of-Thoughts model to reason about this information. Our experiments on a hybrid corpus of simulated and real-world data demonstrate that our framework effectively detects conversational behaviors, generates plausible rationales, and successfully transfers its reasoning

Table 6: Human ratings (1–10) of GoT prediction rationales across audio sources. Higher is better. Rubric: 1–3 unhelpful/incorrect; 4–6 partially correct; 7–8 correct and useful; 9–10 precise and causally grounded. We report de-normalized means across items.

| Audio Source | Mean Rating (1–10) |
|---|---|
| GPT-4 (full-duplex) | **7.07** |
| Moshi (full-duplex) | 6.85 |
| Simulation dataset audio | 6.30 |

capabilities to audio from current dialogue models. We hope this work not only advances the development of interpretable dialogue systems but also inspires further research into the causal structure of human conversation. We will release our code, models, and dataset upon acceptance.

**Limitations.** Our approach, while demonstrating potential, has several important limitations that highlight open research questions. A primary challenge lies in our use of discrete speech-act labels, as real-world conversational cues are often ambiguous and continuous. Future work could explore probabilistic or fuzzy representations to better capture this nuance. Furthermore, the system's performance is sensitive to upstream errors from ASR, and improving its robustness in noisy conditions is a critical next step. Finally, while our synthetic data is effective for training, it may not capture the full diversity of speaking styles, accents, or cultural norms present in human dialogue.

## ETHICS STATEMENT

Our work adheres to the ICLR Code of Ethics. Our goal is to advance full-duplex conversation systems by making the decision process more interpretable via predicting speaker behavior. Our hybrid corpus mixes simulated dialogues with real recordings gathered under license consent, with IRB/ethics review or exemption where applicable. No minors or vulnerable populations were involved. We use deidentified or synthesized speaker data. All experiments were conducted on publicly available datasets, ensuring data privacy and consent. Our research is intended for positive applications, and we do not intend for any surveillance or manipulative purposes.

## REPRODUCIBILITY STATEMENT

To ensure the reproducibility of our results, we will make our source code, including implementation, training, and evaluation scripts, publicly available. All datasets used are public and cited accordingly. The main experimental setup is described in Sec 5.2. For a more detailed breakdown of the model architecture, hyperparameters, and implementation specifics, please refer to Appendix A.3, A.4, andA.5.

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

# A  APPENDIX

## A.1  GoT data processing

For each one-second audio segment, we first extract context nodes from the preceding window using an ASR model (Whisper) OpenIE (Angeli et al., 2015). Next, we apply the speech-acts prediction model's inference method to obtain two speech-acts nodes (one high-level and one low-level) for that segment. We merge these two sets of nodes to form an primary Graph of Thoughts (GoT), which can be represented by an adjacency matrix (Figure 4).

The ground-truth rationale text for the segment is provided in Figure 5. During training, this ground-truth rationale is used as the supervision signal. At inference time, we use the pre-trained model to generate a predicted rationale text for the segment (also shown in Figure 5). We illustrate the encoder's attention weights to provide a more intuitive view of how GoT performs deductive reasoning (Figure 4).

## A.2  Simulation dataset pipeline

We first obtain story narratives from ExploreToM (for use with the LLaMA-3.1-70B-Instruct model). ExploreToM (Sclar et al., 2024b) is a large-scale framework for generating diverse, adversarial theory-of-mind (Brüne & Brüne-Cohrs, 2006) story data, which provides broad coverage of narrative scenarios. These narratives help ensure the plausibility of the stories and the soundness of subsequent rationale reasoning under the GoT framework. We then prompt GPT-4o to convert each narrative into a two-speaker dialogue, using the prompt shown in Table 7. In this task, we require the AI (as the listener) to be proactive about backchannels and interruptions, so we focus on those events and ignore silence.

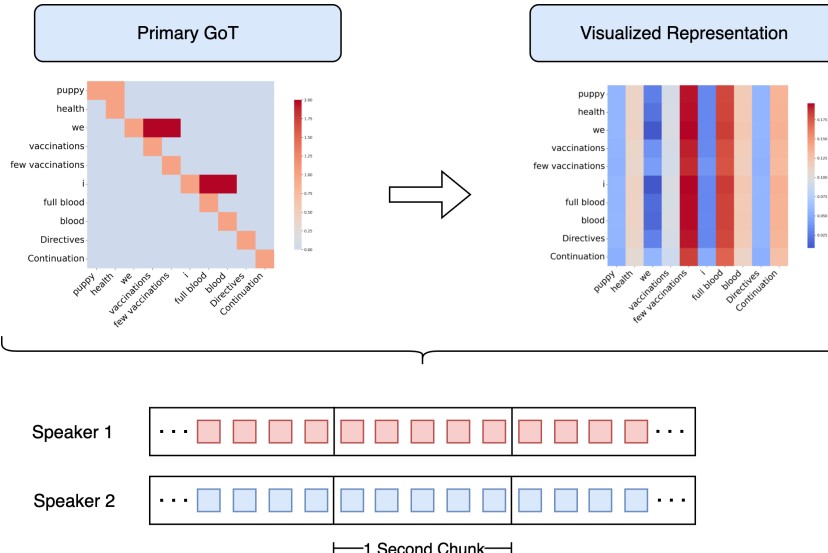

Figure 4: An example primary GoT representation for a one-second audio segment (Appendix A). The left panel shows the adjacency matrix (zeros initially), and the right panel the corresponding visualized representation of encoder's attention weight. Nodes represent context and speech-act concepts.

**Rationale Text** (Ground Truth)

At second 10, speaker1 continues with 'Yeah, and I was also considering a full blood panel, just to' as a directive, indicating a continuation of their plan for the puppy's health checkup. (labels: Directives, Continuation).

**Rationale Text** (Predicted)

At second 10, speaker1 issues a 'Directives' by suggesting to 'start with some basic tests,' marking a 'Continuation' of their plan to address the puppy's health issues.

Figure 5: An example of rationale text ground truth and predicted rationale generated by our GoT model. For overall accuracy statistics, see Sec 6.2

Next, we generate audio for each dialogue using CosyVoice2's (Du et al., 2024) `inference_instruct2` engine. Based on our experiments, TTS output is more stable when provided with a reference audio (i.e., conditioning on a sample utterance). Therefore, for each generated utterance, we select a single-speaker recording (from a separate dataset) to serve as a timbre/style reference; this reference is *not* part of the text prompt. Our pipeline introduces a novel overlap-based dialogue-stitching mechanism: we first synthesize each utterance independently with TTS, then use the [backchannel] and [interruption] markers to compute timestamps and insert each subsequent utterance into the previous one at the corresponding time, ensuring overlap for interruptions and backchannels. Finally, we place the first speaker's audio on the left channel and the second speaker's audio on the right channel, yielding a two-channel overlapped speech dataset.

With this design, our dataset emphasizes overlapping dialogue events. Compared to human–human conversations in the Talking-Turns (Arora et al., 2025a) benchmark (see Table 8), our simulation has a higher proportion of backchannels and interruptions and avoids extreme class imbalance. This helps mitigate class-imbalance issues when training the speech-acts prediction model. Simultane-

---

**Task:** Write a natural dialogue between two speakers and label each utterance with one behavior tag.
**Rules:**

- Speakers: exactly two.
- Use the provided {narrative}.
- Length: 5–8 utterances total.
- Include at least one interruption event and one backchannel event.
    - **Interruption:** put "[interruption]" inside the cut-off utterance. Mark the interrupter by adding "(interruption)" after their name on the next line.
    - **Backchannel:** create a separate backchannel utterance (1–3 words) and mark that speaker with "(backchannel)"; also insert "[backchannel]" inside the other speaker's ongoing utterance at the exact insert point.
- After every utterance, append exactly **one** tag in braces from {Constatives, Directives, Commissives, Acknowledgments}.

**Narrative:** {narrative}
**Output Format:**

(1) speaker1: utterance {Intent}

(2) speaker2: utterance {Intent}

(i) speaker1: ... [interruption] ... {Intent}

(i+1) speaker2(interruption): ... {Intent}

(j) speaker1: ... [backchannel] ... {Intent}

(j+1) speaker2(backchannel): ... {Intent}

...

(N) speakerX: utterance {Intent}

---

Table 7: Prompt template for generating simulated dialogues from each narrative. Numbers in parentheses denote utterance indices; speaker names and utterances are labeled with an intent tag.

ously, we generate per-second labels for both high-level and low-level speech acts. For the low-level label, we assign priorities: backchannel > interruption > turn-taking > continuation. For the high-level label, we use the {Intent} tag of the corresponding utterance.

|  | (a) % Turn Change | (b) % Backchannel | (c) % Interruption |
|---|---|---|---|
| Human Ref (in-domain) | 15.9 | 0.30 | 0.40 |
| Speech-Acts Simulation Data | 11.4 | 6.4 | 17.6 |

Table 8: Event distribution (%) in dialogue: (a) turn changes, (b) backchannels, (c) interruptions. "Human Ref" is the in-domain human conversation reference; "Simulation" is our generated dataset.

## A.3 SPEECH ACTS PREDICTION MODEL

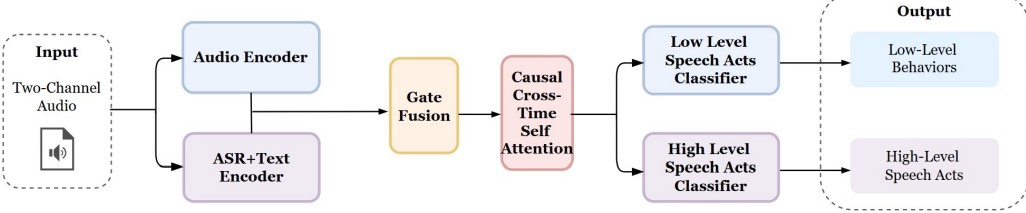

Figure 6: Speech Acts Prediction Model framework

### A.3.1 INPUT AND SEGMENTATION.

Let $X = \{x_t\}_{t=1}^T$ be a single–channel waveform at 16kHz. We split $X$ into a sequence of non–overlapping 1-second chunks $U = \{U_i\}_{i=1}^N$ with $U_i = x_{B_{i-1}+1:B_i}$ and $B_i = i \cdot N_{\text{block}}$, where $N_{\text{block}}$ is the number of samples per second. The model is *causal*: it predicts the speech–act labels for chunk $i$ using only past context $U_{1:i-1}$.

### A.3.2 PREDICTION OBJECTIVE.

We estimate the label sequence $L = \{(l_i^{\text{hi}}, l_i^{\text{lo}})\}_{i=1}^N$ by maximizing the posterior

$$P(L \mid X) = \prod_{i=1}^N P\big(l_i^{\text{hi}}, l_i^{\text{lo}} \mid U_{1:i-1}\big). \tag{4}$$

Under a conditional–independence approximation of the two heads given semantic context,

$$P\big(l_i^{\text{hi}}, l_i^{\text{lo}} \mid U_{1:i-1}\big) \approx P\Big(l_i^{\text{hi}} \mid \hat{U}_i\Big) \cdot P\Big(l_i^{\text{lo}} \mid \hat{U}_i\Big), \tag{5}$$

where we further condition only on a fixed–length causal window $\hat{U}_i = x_{(B_{i-1}-W):B_{i-1}}$ of size $W$:

$$P(L \mid X) \approx \prod_{i=1}^N P\Big(l_i^{\text{hi}} \mid \hat{U}_i\Big) \; P\Big(l_i^{\text{lo}} \mid \hat{U}_i\Big). \tag{6}$$

### A.3.3 SPEECH & TEXT ENCODERS.

Each chunk $U_i$ is encoded by two frozen off–the–shelf encoders: a HuBERT speech encoder producing $h_i^{\text{B}} \in \mathbb{R}^{768}$ and an ASR→Text stack that first transcribes $U_i$ with Whisper and then encodes the transcript by a T5 encoder to yield $h_i^{\text{E}} \in \mathbb{R}^{768}$ (layer weights in both stacks are fixed). The two streams are fused by a gated residual:

$$\lambda_i = \sigma\big(W_{\text{B}}h_i^{\text{B}} + W_{\text{E}}h_i^{\text{E}}\big), \quad h_i = (1 - \lambda_i)\, h_i^{\text{B}} + \lambda_i\, h_i^{\text{E}}. \tag{7}$$

### A.3.4 CAUSAL TEMPORAL ENCODER.

We apply a causal Transformer over time on $\{h_1, \ldots, h_i\}$ with a standard triangular mask that forbids access to future chunks, and take the $i$-th output as contextual state $z_i$:

$$z_i = \text{Transf}_{\text{causal}}(h_{1:i})[i]. \tag{8}$$

### A.3.5 CLASSIFICATION HEADS AND TRAINING LOSS.

Two linear heads followed by softmax produce the distributions

$$p_i^{\text{hi}} = \text{Softmax}\big(W^{\text{hi}}z_i\big), \qquad p_i^{\text{lo}} = \text{Softmax}\big(W^{\text{lo}}z_i\big). \tag{9}$$

The high–level head is 4-way (task–specific taxonomy), and the low–level head is *also* 4-way over speech–act events {TURNCHANGE, INTERRUPTION, BACKCHANNEL, CONTINUATION} (one label per second). We optimize the sum of weighted cross–entropy losses with standard padding masks:

$$\mathcal{L} = \sum_{i=1}^N \Big[\alpha \cdot \text{CE}(y_i^{\text{hi}}, p_i^{\text{hi}}) + \beta \cdot \text{CE}(y_i^{\text{lo}}, p_i^{\text{lo}})\Big], \tag{10}$$

where class–imbalance is handled by inverse–frequency weights absorbed into coefficients $(\alpha, \beta)$ per class, and masked positions use the ignore index.

### A.3.6 OPTIMIZATION AND RUNTIME.

Unless otherwise noted, we use AdamW (lr $= 3 \times 10^{-4}$, weight decay $= 10^{-2}$) with a linear warmup of 1,000 steps, batch size 8, and 10 epochs. Training is mixed–precision (AMP) and data–parallel via DDP. Encoders are frozen; only the fusion module (equation 7), the causal Transformer, and both classification heads are updated. Inference supports both offline and streaming (online–cached) modes with the same causal window $\hat{U}_i$ and one–second latency per step.

### A.3.7 Output format.

For each chunk $i$, the model emits $(\hat{l}_i^{\text{hi}}, \hat{l}_i^{\text{lo}})$ along with per–class posterior vectors. The low–level head strictly outputs the four speech–act categories above; high–level outputs follow the 4-class taxonomy used in our datasets. All labels align exactly to the 1 s grid used by the segmentation.

## A.4 GoT reference generation Model

### A.4.1 Problem setup and causality.

Given a single–channel conversation waveform $X = \{x_t\}_{t=1}^T$ at 16kHz, we form a sequence of non–overlapping 1 s decision steps and, for each step $i$, condition only on a fixed left context window $\hat{U}_i = x_{(B_{i-1}-W):B_{i-1}}$ with $B_i = i \cdot N_{\text{block}}$ (samples per second $N_{\text{block}}$; default $W{=}30\text{s}$). The goal is to generate a *reference rationale* $r_i$ (a short natural–language explanation of what happens around second $i$) using only past information. Under this causal assumption we maximize

$$P(R \mid X) = \prod_{i=1}^N P\Big(r_i \mid \hat{U}_i, \mathcal{G}_i\Big), \tag{11}$$

where $\mathcal{G}_i = (\mathcal{V}_i, \mathcal{E}_i)$ is a symbolic *Graph-of-Thought* context extracted from the same window (defined next).

### A.4.2 Windowed ASR and textual context.

We run an incremental ASR on $\hat{U}_i$ (Whisper–style backend; word timestamps on) to obtain the cumulative text visible at second $i$, denoted $S_i$. We keep only words whose end–time $\leq W$ to respect causality. This $S_i$ is the sole textual evidence later consumed by the text encoder and by the IE step below.

### A.4.3 OpenIE → content graph.

From $S_i$ we apply OpenIE (CoreNLP) to extract predicate–argument triples and collapse them into a node list $\mathcal{V}_i^{\text{text}} = \{\text{span}_k\}$ and an undirected adjacency $\mathbf{A}_i^{\text{text}} \in \{0,1\}^{|\mathcal{V}_i^{\text{text}}| \times |\mathcal{V}_i^{\text{text}}|}$ (self–loops on the diagonal). Nodes are unique surface spans (deduplicated by string match), and edges mark co–occurrence within the same triple.

### A.4.4 Speech–act augmentation.

In parallel we run the supervised speech–acts model on the raw audio to obtain per–second low/high labels $(\ell_i^{\text{lo}}, \ell_i^{\text{hi}})$ (see App. A.3). We append two categorical nodes that summarize these labels,

$$\mathcal{V}_i^{\text{sa}} = \big\{ \texttt{SA\_High=}\ell_i^{\text{hi}}, \ \texttt{SA\_Low=}\ell_i^{\text{lo}} \big\},$$

and expand the adjacency to $\mathbf{A}_i$ by block–diagonal padding with identity on the new nodes (link edge ). The final GoT context is $\mathcal{G}_i = (\mathcal{V}_i, \mathbf{A}_i)$ with $\mathcal{V}_i = \mathcal{V}_i^{\text{text}} \cup \mathcal{V}_i^{\text{sa}}$.

### A.4.5 Multi–modal encoders and fusion.

Each window forms a triplet $(\hat{U}_i, S_i, \mathcal{G}_i)$. (i) *Audio encoder:* a frozen HuBERT stack yields $h_i^{\text{a}} \in \mathbb{R}^d$ from $\hat{U}_i$ (last frame of the window). (ii) *Text encoder:* a frozen T5 encoder yields $h_i^{\text{t}} \in \mathbb{R}^d$ from $S_i$ (layer–weighted pooling). (iii) *Graph encoder:* a GAT processes $(\mathcal{V}_i, \mathbf{A}_i)$ to produce $h_i^{\text{g}} \in \mathbb{R}^d$ (mean over node embeddings). We compute a gated residual fusion

$$\lambda_i = \sigma(W_{\text{a}} h_i^{\text{a}} + W_{\text{t}} h_i^{\text{t}} + W_{\text{g}} h_i^{\text{g}}), \quad h_i = (1 - \lambda_i) \, h_i^{\text{a}} + \lambda_i \, (h_i^{\text{t}} + h_i^{\text{g}}), \tag{12}$$

and pass $\{h_1, \ldots, h_i\}$ through a causal Transformer with a triangular mask to obtain the contextual state $z_i$.

Table 9: Encoder combinations

| Combination | B-1 | R-1 | R-L | SIM | PARAM |
|---|---|---|---|---|---|
| HuBERTlarge + T5base | 0.42 | 0.31 | 0.35 | 0.46 | 791.6M |
| HuBERTbase + T5large | 0.54 | 0.52 | 0.48 | 0.57 | 1.1B |
| HuBERTlarge + T5large | 0.50 | 0.48 | 0.43 | 0.52 | 1.3B |

### A.4.6 SEQ2SEQ GENERATION AND TRAINING.

A T5 decoder conditions on $z_i$ to generate the rationale tokens $r_i = \{y_{i,1}, \ldots, y_{i,T_i}\}$ with the standard left–to–right factorization

$$P(r_i \mid \hat{U}_i, \mathcal{G}_i) = \prod_{t=1}^{T_i} P(y_{i,t} \mid y_{i,<t}, z_i).\tag{13}$$

Training uses teacher forcing with cross–entropy over the target rationales (max length 96 tokens), masking padded positions. All encoders are frozen; we update the fusion (12), the causal Transformer, the graph encoder, and the decoder. Optimization follows App. A.3 (AdamW, warmup, AMP, DDP).

### A.4.7 PREPROCESSING & ARTIFACTS.

For each audio we emit, at every second $i$, a JSON record with fields `audio_id`, $i$, `text_window=` $S_i$, `nodes_path`, `adj_path`, `rationale_gt`. Per-audio indices are concatenated into a corpus index `preproc_index.jsonl`. Inference reuses the same causal window and writes one JSONL with fields `audio_id`, `t`, and `rationale`; evaluation aligns predictions with ground truth by keys (`audio_id, t`) and reports BLEU/ROUGE metrics.

## A.5 TRAINING DETAILS

We train models using AdamW (Loshchilov & Hutter, 2019) with learning rate $3 \times 10^{-4}$ and weight decay $10^{-2}$. The learning rate follows a linear decay schedule with 1,000 warm-up steps. We use automatic mixed precision (fp16) and clip gradient norms at 1.0. Training is done with a batch size of 8 per GPU (no accumulation) for 10 epochs, with a checkpoint saved each epoch and final evaluation on the last epoch. We use PyTorch and DistributedDataParallel (Li et al., 2020) when multiple GPUs are available (otherwise we train on a single 48-GB GPU).

For the speech-acts predictor, encoder weights are initialized from public pretrained checkpoints and kept frozen; only the task-specific classification heads are trained. Audio is resampled to 16 kHz and segmented into contiguous 1-second frames, with frame-level supervision spanning each utterance. To address class imbalance, we apply a class-reweighted cross-entropy loss (Polat et al., 2025) using inverse-frequency weights computed from the training split. Training a single run on one high-memory GPU takes about 48 hours.

For the GoT rationale generator, each example uses a strictly causal audio window $(t - W, t]$ with $W = 30\,\mathrm{s}$ (no lookahead), at 16 kHz. Text inputs are tokenized with the T5 tokenizer , and graph inputs (node features and adjacency) are padded to fixed size per batch. We use the same AdamW schedule as above (same batch size and epochs). During sequence generation we apply teacher forcing for stability. A single run takes about 7 hours on one 48-GB GPU. By default, we fix the random seed to 42. For results reported as mean $\pm$ std, we perform at least five independent runs with different seeds and compute statistics over those runs.

## A.6 ENGINEERING METRICS

Referencing the encoder combinations in Table 9, the decoder scale (T5) is the main lever for quality versus cost. The `HuBERTbase + T5large` configuration (1.1B parameters) achieves optimal performance across most metrics: BLEU-1 (0.54), ROUGE-1 (0.52), ROUGE-L (0.48), and SIMILARITY (0.57). Counterintuitively, this mid-scale approach outperforms the largest

Table 10: Window size (**W**) × Look ahead (**L**) vs. metrics (**A+T+GAT** *only*)

| L | W | A+T+GAT | | | |
|---|---|---|---|---|---|
| | | **B-1** | **R-1** | **R-L** | **SIM** |
| **0** | 10 | 0.2829±0.0031 | 0.2626±0.0037 | 0.2136±0.0028 | 0.3178±0.0055 |
| | 20 | 0.2863±0.0065 | 0.2669±0.0079 | 0.2187±0.0078 | 0.3220±0.0111 |
| | 30 | 0.2899±0.0066 | 0.2709±0.0083 | 0.2216±0.0069 | 0.3299±0.0101 |
| | 40 | 0.2899±0.0066 | 0.2709±0.0083 | 0.2216±0.0069 | 0.3299±0.0101 |
| **5** | 10 | 0.3197±0.0073 | 0.3005±0.0073 | 0.2463±0.0054 | 0.3569±0.0067 |
| | 20 | 0.3203±0.0049 | 0.3037±0.0056 | 0.2497±0.0046 | 0.3615±0.0057 |
| | 30 | 0.3246±0.0072 | 0.3081±0.0070 | 0.2515±0.0067 | 0.3665±0.0084 |
| | 40 | 0.3246±0.0072 | 0.3081±0.0070 | 0.2515±0.0067 | 0.3665±0.0084 |
| **10** | 10 | 0.4121±0.0082 | 0.3988±0.0080 | 0.3485±0.0099 | 0.4519±0.0090 |
| | 20 | 0.4042±0.0047 | 0.3897±0.0059 | 0.3434±0.0047 | 0.4401±0.0092 |
| | 30 | 0.4094±0.0040 | 0.3948±0.0048 | 0.3477±0.0040 | 0.4453±0.0064 |
| | 40 | 0.4094±0.0040 | 0.3948±0.0048 | 0.3477±0.0040 | 0.4453±0.0064 |

*A+T+GAT* = Audio + Text + Graph (GAT).

`HuBERTlarge` + `T5large` configuration (1.3B parameters), which scores lower, indicating diminishing returns when scaling both components simultaneously.

Upgrading from T5-base to T5-large consistently improves all four text metrics, with the trade-off lying in increased decoding latency and memory. Empirically, decoding time increases by about 1.6–2.0×, but the quality-per-compute gain is strongest, especially for ROUGEL (Lin, 2004) and SIMILARITY (Reimers & Gurevych, 2019). Whisper upgrades matter most in noisy settings: moving from small to large chiefly improves BLEU1/ROUGE1/SIMILARITY (fewer ASR errors → higher lexical/semantic alignment) at a 2–4× cost on the ASR side; gains diminish on clean audio. HuBERT (Hsu et al., 2021) upgrades mainly support coherence: base to large yields steady gains in ROUGEL/SIMILARITY at a 1.2–1.5× feature-side cost—an inexpensive, reliable boost.

Across choices, modality > future peeking > blindly extending the window: under strict causality (lookahead = 0), investing budget in audio+text+GAT offers better value than increasing lookahead or window length, consistent with the causal analysis in Section 5.3. The performance plateau between 1.1B and 1.3B parameters suggests that computational resources beyond the mid-scale configuration should be allocated to other optimizations rather than further encoder scaling. For low-latency, compute-constrained deployments, `HuBERT-large` + `T5-large` + `Whisper-small` provides a good quality/latency balance, especially with $W = 20$–$30$ s and $L = 0$.

## A.7 CAUSAL AND MULTIMODAL FUSION

**Streaming Setup.** We operate in a *strictly causal* regime: at each 1 s hop $t$, the model conditions only on the most recent $W$ seconds of audio $(t$–$W, t]$ and, optionally, a small look-ahead $L$ admitting information available by $t$+$L$.

**Findings.** The grid in Fig. 3 and Table 10 ($W \in \{10, 20, 30, 40\}$, $L \in \{0, 5, 10\}$, modalities $A$, $A+T$, $A+T+$GAT) yields three consistent results:

(i) Look-ahead dominates. Increasing $L$ from 0 to 10 improves audio-only by $\approx +0.14$ absolute on BLEU-1/ROUGE-1/ROUGE-L/SIM (averaged over $W$, indicating a small, deployable $L$ stabilizes tokenization without violating causality.

(ii) A window $W \in [20, 30]$ is most effective across metrics. Shorter windows lack context while longer ones dilute recency with stale evidence.

(iii) Under streaming, naïvely adding text hurts. $A+T$ underperforms $A$ by $\approx 0.02$ on average, while incorporating the GoT graph yields a small, consistent recovery over $A+T$ (+0.002–+0.006) yet remains below audio-only. We attribute this to noise from partial ASR and imperfect real-time graph extraction; the causal gate downweights text streams when confidence is low. In practice,

we adopt $W = 30$ and $L = 10$ with conservative gating, offering the best latency–accuracy trade-off (reported as mean±sd with 95% CIs in Table 10).

**Takeaways.** Under strict causality and tight latency, the *speech stream is the reliable backbone* while text streams help only when their *reliability and temporal consistency* are controlled. A small $L$ provides a practical stability gain, and the optimal $W$ is governed by freshness of evidence rather than sheer length. These observations motivate *opportunistic multimodal fusion*: anchor decisions in audio, admit text/graph only when confidence is high, and regulate their influence via causal masking, temporal decay, and confidence-aware gating. This preserves streaming constraints, mitigates noise-driven failures, and enables adaptive $L/W$ policies with incremental, confidence-weighted graph construction.

### A.8 HUMAN STUDY ON GOT RATIONALES FOR FULL-DUPLEX AUDIO FMS

**Models and setting.** We assess whether **GoT's prediction rationales** are *plausible and action-guiding* when applied to full-duplex conversational audio from **GPT-4**, **Moshi**, and our **simulation** corpus. The human study is explicitly scoped to **content-level reasoning**: raters judge (i) whether the rationale identifies the correct **speech act**, (ii) whether it infers a sensible **speaker intent**, and (iii) whether the **discourse logic** that links local evidence to the predicted action is coherent. We do **not** require, evaluate, or match the underlying **distribution of dialogue events** (e.g., interruption or backchannel frequencies) for any source; the comparison is strictly about the **reasonableness of GoT's explanations** given the audio context.

Under this rubric, the corpus-level means (1–10 scale) are:

$$\textbf{Simulation} = \textbf{8.93} \ > \ \textbf{GPT-4} = \textbf{7.06} \ > \ \textbf{Moshi} = \textbf{4.25} \,.$$

Three qualitative patterns align with these scores. *(1) Simulation* achieves the top score even though rationales are often **more abstract** in phrasing: training extracts *higher-level regularities*, so explanations read rule-like yet **consistently hit the correct speech act and polarity** (affirmation/negation). They **stabilize early** (from $\approx 3\,\text{s}$) with very low error and provide **strong short-horizon forecasting**, aided by our *prediction model*. *GPT-4* yields **more concrete, locally grounded** rationales but with a **slightly higher error rate** than simulation—still squarely in the "correct and useful" band. *(3) Moshi* trails due to **omitting key local cues** more frequently even though equipped with stronger turn-taking capacity compared with **GPT-4**; raters note **greater temporal volatility**, with coherence changing markedly as the clip progresses.

**Inference based on scores.** The **simulation** advantage reflects a trade-off: its rationales are *less colloquial but more systematically correct*, especially for fine-grained polarity and moment-level act detection, and they remain **stable after the first few seconds**. By contrast, **GPT-4**'s rationales are *rich in concrete references* (prosody, pause structure, partner state) yet admit **slightly more slips** than simulation. **Moshi**'s lower score is consistent with **evidence omission and drift**: salient turn-management signals are not always surfaced, and the internal logic of the explanation can **shift substantially over time**.

**Transferability.** Despite being trained on *simulated* dialogues, GoT's rationales **transfer well** to real full-duplex audio: performance on **GPT-4** remains strong (7.06), while **Moshi** lags (4.25). The pattern suggests that GoT captures **domain-stable timing features** (silence windows, overlap context, prosodic shifts) that support explanation on real model audio without domain-specific fine-tuning, while also benefiting from the **high-coverage abstraction** learned in simulation. Practically, this endorses a two-step recipe: *pre-train on simulation for robust, early-stabilizing predictions and accurate polarity judgments; then validate and calibrate on real model audio* to increase concreteness and reduce residual errors, particularly for systems like Moshi where **key-cue omission and temporal variability** are more pronounced.

### A.9 SIMULATION DATASET QUALITY

#### A.9.1 TURN-TAKING CORPUS STATISTICS

We first briefly explain the terminologies: an **IPU** is a continuous stretch of speech on one channel delimited by $> 200\,\text{ms}$ silence on both sides; **silence** splits into **Pause** (same speaker) vs **Gap**

(across speakers); **Overlap** denotes simultaneous voice activity on both channels. Short within-turn overlaps function as backchannels; longer overlaps indicate interruptions.

**Counts per minute.** Our corpus exhibits IPU 23.06 /min, Pause 10.7 /min, Gap 7.3 /min, Overlap 6.7 /min. Relative to the human reference (15.7, 3.8, 5.5, 6.6), we observe denser micro-segmentation: more IPUs and more within-speaker Pauses per minute, while Gap and Overlap rates are closer to human. This pattern suggests shorter, more frequent IPUs—likely driven by abundant backchannels or clause-internal hesitations—rather than longer, monolithic turns, consistent with established observations of conversational floor management (Sacks et al., 1974).

**Cumulated duration.** Our durations (percent of conversation time) are IPU 84.7%, Pause 9.6%, Gap 1.6%, Overlap 4.2%, whereas the human reference shows 97.3%, 5.7%, 3.7%, 6.7%. Thus, while we count overlaps nearly as often as humans, their mean length is shorter (4.2% vs. 6.7%), and we spend more time in within-speaker pauses and less in cross-speaker gaps. Practically, the corpus has more micro-breaks within a turn and less turn-exchange silence, which is consistent with frequent short acknowledgments that do not wrest the floor, a phenomenon also highlighted in cross-linguistic studies of overlap and backchanneling (Stivers et al., 2009).

Conclusions.

1. The high IPU/min + high Pause/min combination points to fine-grained intra-turn rhythm (listeners or speakers insert many brief hesitations).
2. Overlap/min $\approx$ human but shorter overlaps indicates backchannel-style micro-overlaps rather than long interruptions—useful for interactive feel but currently a bit under-energized compared to human dialogues.

These interpretations align with how recent evaluation work distinguishes global distributions (counts/durations) from the timing quality of turn-taking behavior.

### A.9.2 ACOUSTIC QUALITY AND INTERACTION COUPLING

Beyond raw distributions, we examine whether acoustic quality correlates with turn-management behavior (e.g., SNR quintiles vs Gap%, Overlap CPM vs $F_0$ variance). The theory is simple: lower SNR depresses the system's confidence in floor-transfer cues, inflating gaps; higher prosodic variability ($F_0$/RMS var.) invites micro-overlaps/backchannels.

These analyses connect the conditions of the channel to interactional outcomes—a relationship emphasized in recent timing-centric evaluations, where choices like when to speak up or when to backchannel are modeled explicitly rather than inferred only from global rates.

### A.9.3 SENTENCE LENGTH STATISTICS

At the text layer, our simulated dialogues have mean sentence length 10.11 tokens ($\sigma = 1.29$; p10/p50/p90 = 8.61/10.00/11.71). This narrow band supports the picture painted by the audio-side metrics: short IPUs and frequent within-turn breaks rather than long, syntactically heavy turns. For downstream modeling, this implies dense turn opportunities and more frequent floor-keeping cues to resolve.

Table 11: Sentence length statistics of the simulated dialogues.

|                          | Mean  | Std  | P10  | P50   | P90   |
|--------------------------|-------|------|------|-------|-------|
| Sentence length (tokens) | 10.11 | 1.29 | 8.61 | 10.00 | 11.71 |

### A.9.4 NOISE AND OVERLAP RATIOS

The ECDF[1] indicates a broad SNR coverage (p10 $\approx$ 23 dB, p50 $\approx$ 33 dB, p90 $\approx$ 41 dB), with a modest low-SNR tail. The corpus is neither overly clean nor dominated by noisy outliers. Overlap ratios are strongly left-skewed (mode $< 2\%$, long tail up to $\approx 22\%$), consistent with brief, supportive overlaps rather than sustained interruptions. Noise primarily affects gap formation rather than overlap intensity, while overlap appears more tied to prosodic activity; we therefore stratify evaluation by SNR bins and supplement high-overlap slices for robustness.

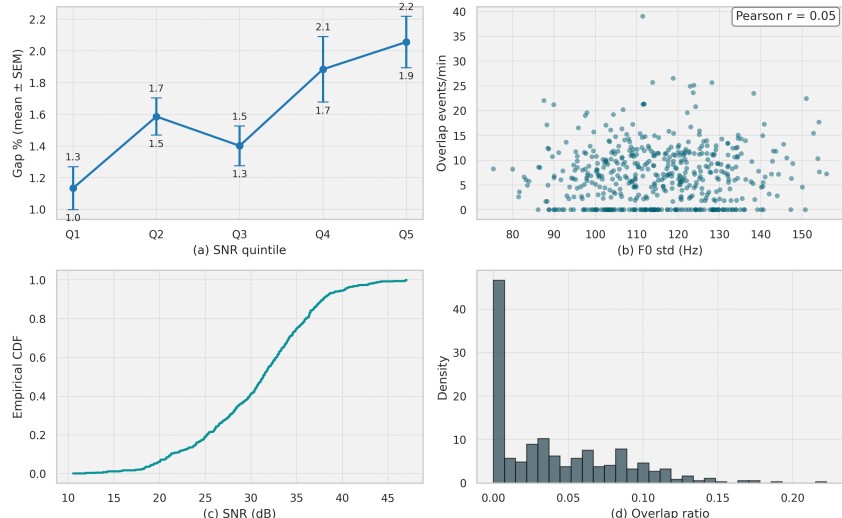

Figure 7: (a) SNR cumulative distribution (ECDF), (b) Overlap ratio distribution, (c) Gap percentage by SNR quantile, (d) Overlap count per minute vs. $F_0$ variance.

### A.9.5 NATURALNESS

Our results paint a picture of interactive yet micro-segmented exchanges. The system produces a human-like rate of overlaps but those overlaps are shorter on average, while both IPU/min and Pause/min are elevated. In conversational terms, this combination signals many brief acknowledgments, clause-internal hesitations, and light cues that sustain the current floor, rather than the longer cooperative overlaps humans often use to co-construct content (Schegloff, 1982). The effect is a dialogue that feels responsive and attentive, yet parcels speech into smaller units than natural talk would, creating a fine-grained turn rhythm with frequent re-entry points.

The speaking style metrics reinforce this reading. Higher WPM together with a higher filler-word rate indicates fast, hedge-rich delivery: the system keeps the channel active and signals stance ("uh-huh", "yeah", "okay") while avoiding hard floor grabs. In duplex settings this is advantageous—short backchannels and brief pauses allow the model to juggle listening and speaking without starving the partner—but it also compresses turns and can make the discourse feel punctuated rather than smoothly interleaved. From a control perspective, fillers operate as a low-cost mechanism for grounding and timing calibration; they maintain engagement without triggering a full turn transition, a role consistent with prior findings on backchannels as coordination signals (Yngve, 1970).

These observations motivate timing-centric evaluation beyond corpus totals. Global distributions reveal how much of each event type we produce, but interaction quality is governed by when speak-up, backchannel, and interruption actions are taken. A system can match human totals while still sounding mechanical if the micro-decisions are misplaced by a few hundred milliseconds. Judge-based protocols at frame-level resolution (e.g., 40 ms decisions) directly evaluate the hazard of taking the floor given local cues—silence duration, prosody, and partner state—and let us verify that our short overlaps behave as supportive backchannels rather than butting in. In practice this amounts to checking conditional timing curves (e.g., the probability of speaking as a function of recent silence or pitch movement) and calibrating thresholds so that the induced dwell-time distributions (for overlaps, pauses, and gaps) match human ranges.

Putting it together, our corpus is fast, hedge-rich, and highly segmented, with short supportive overlaps and many within-turn micro-pauses. To move closer to human conversational texture, we should slightly lengthen supportive overlaps (nudging their dwell time upward without increasing interruption rate) and smooth within-turn micro-breaks so IPUs aggregate into more natural chunks. The recommended next step is to couple these policy adjustments with a judge model that scores tim-

---

[1]ECDF: empirical cumulative distribution function; SNR: signal-to-noise ratio; CPM: counts per minute; $F_0$: fundamental frequency.

ing decisions in context; passing that test would confirm that our short overlaps function as intended—lightweight grounding signals that enhance, rather than disrupt, the flow of conversation.

## A.10 THE USE OF LARGE LANGUAGE MODELS

During the writing of this paper, we leveraged GPT-5 to aid in polishing the writing. Its use was exclusively limited to improving grammar, phrasing, and overall readability. The core scientific contributions, experimental results, and intellectual content are entirely our own.

