# OpenReview forum: "Enabling Conversational Behavior Reasoning Capabilities in Full-Duplex Speech"
_ICLR.cc/2026/Conference — ICLR 2026 Conference Withdrawn Submission_

### Official Review · Reviewer_cFJ8 · 2025-10-26

**Soundness:** 2
**Presentation:** 1
**Contribution:** 2
**Rating:** 2
**Confidence:** 3

**Summary:**

The paper presents a Graph-of-Thought (GoT) framework for full-duplex spoken dialogue systems that aims to mimic human reasoning rather than simply predict tokens. It identifies conversational behaviors at both high (intent) and low (speech act) levels, using causal reasoning to anticipate and explain the next action in a conversation. Trained on a mix of simulated and real dialogues enriched with human-provided rationales, the system produces interpretable chains of reasoning behind its conversational decisions.

**Strengths:**

Incorporating comprehensive reasoning in duplex systems is challenging due to its latency issues. While many previous works focus on low-level signal-based evaluation, this paper properly argues the importance of addressing cognitive and high-level features.

**Weaknesses:**

1. **Data synthesis**:
    - The paper needs to provide a more detailed and systematic description of the data synthesis pipeline, including how reproducibility and verification are ensured. Several key implementation details are omitted, for instance, how candidate backchannels are generated (e.g., whether they rely entirely on GPT prompts). The statement "we deliberately introduce controlled overlap between speakers" is also ambiguous and should be clarified. Evaluation on unverified datasets does not substantially strengthen the contribution to the community. Lines 260-261 suggest possible modifications to datasets for improved training (i.e., intentionally increasing events even though natural dialogues are actually not); if this is the case, it raises a question of whether the resulting training objective still reflects the true data distribution.

1. **Evaluation metric**:

    - Latency is a critical factor when evaluating real-time duplex systems, yet it is not reported in the current version. Additional reasoning modules within a pipeline can increase latency, and it is important to analyze this trade-off. Moreover, generation accuracy may not be the most appropriate metric in this context, and syntactic metrics such as BLEU or ROUGE may not adequately capture the semantic quality of generated rationales. A further clarification is requested regarding the measurement of filler word rate: are all types of filler words predefined, or are they dynamically identified?

1. **Reproducibility & Scalability**:
    - Line 311 mentions reliance on human-annotated rationales, but relevant implementation details are lacking. It would be helpful to clarify whether the proposed architecture can be scaled without such annotations, or if human-labeled rationales are always required for effective use.

1. **Self-containedness and readability**: Although some details are provided in the appendices, they are not adequately referenced in the main text, which affects readability and self-containedness. Specific points include:
    - Line 181: The issue of class imbalance is not addressed.
    - Formulations: Definitions are incomplete and should be accompanied by task-specific examples.
        - Line 208: The meaning of "standard basis" should be clarified.
        - Variables (e.g., $v$ in line 208, sa-h & sa-l in line 212) should be defined before use.
    - Acronyms such as IPU should be defined upon first appearance.
    - Baseline results (e.g., Table 5) appear to be missing.
    - References to relevant appendices (e.g., Section A.2 in Section 5.1) should be added to guide the reader.

1. **Editorial quality**:
    - Numerous minor editorial issues were observed, including inconsistencies in capitalization, spacing, and citation formats. Additionally, the ordering of tables (e.g., Table 4 preceding Table 3) should be corrected.

The paper presents an interesting idea but requires major revisions in methodology, evaluation, and presentation. I recommend rejection in the current cycle, with encouragement to resubmit after substantial improvements.

**Questions:**

See weaknesses.

**Details Of Ethics Concerns:**

Detailed information regarding human annotations should be included. For instance, the manuscript does not specify how the authors recruited or instructed the human annotators.

---

### Official Review · Reviewer_Pgmf · 2025-10-29

**Soundness:** 2
**Presentation:** 3
**Contribution:** 3
**Rating:** 4
**Confidence:** 2

**Summary:**

This paper proposes a novel framework for a full-duplex speech dialogue system that moves beyond direct sequence prediction, instead employing an explicit two-stage process of "next-action awareness and prediction." The core idea is to model the causal reasoning underlying human conversation. The first stage of the system is a hierarchical dialogue act detector that identifies high-level communicative intentions (e.g., declarative intentions, directive intentions) and low-level speech interaction mechanisms (e.g., turn-taking, feedback, interruption) from streaming audio on a second-by-second basis. The second stage uses a Graph-of-Thoughts (GoT) reasoning module, which constructs a dynamic causal graph based on the detected actions and semantic triples extracted from text transcripts. The GoT module then generates natural language explanations that elucidate the predicted next action. To achieve this, the authors created a large-scale (192 hours) synthetic dialogue corpus containing controlled conversational events and manually verified rationales, and validated their method on both this synthetic data and the real-world Candor dataset.

**Strengths:**

Original technical contribution of reframing the full-duplex challenge from a black-box prediction task (next segment/token) to an explicit, interpretable reasoning task (perceive -> reason -> act). Also, the application of a Graph-of-Thoughts (GoT) framework to model the evolving conversational state is a novel and well-motivated architectural choice.​

The authors have developed and released a substantial new dataset, complete with a detailed analysis of its statistical properties compared to human dialogue. The evaluation on both synthetic and real data (Candor) provides a solid empirical grounding for the paper's claims.​

The paper is well-written, clearly motivated, and easy to follow.

**Weaknesses:**

The framework's representation of conversation is coarse, both temporally and semantically. Quantizing the dialogue into one-second chunks and assigning a single discrete speech-act label per chunk oversimplifies the fluid and often ambiguous nature of human interaction, a limitation the authors acknowledge.

**Questions:**

The framework stops at generating reasoning results. How could we further utilize this output  by a full dialogue agent to control its   response? For instance, how would a rationale like "Speaker1 issues a Directive, so Speaker2 should not interrupt" be translated into a precise timing decision (e.g., inhibit speech for the next X milliseconds)?

---

### Official Review · Reviewer_VHMS · 2025-10-31

**Soundness:** 2
**Presentation:** 3
**Contribution:** 2
**Rating:** 2
**Confidence:** 3

**Summary:**

This paper addresses interpretability in full-duplex dialogue systems by proposing a shift from direct token prediction to explicit behavior reasoning. The authors present a two-stage framework consisting of hierarchical behavior detection and Graph-of-Thoughts (GoT) based reasoning that generates natural language rationales.

**Strengths:**

The paper introduces a clear conceptual distinction between pattern matching and reasoning in dialogue systems. The proposed "Perception → Reasoning → Generation" framework provides structure to the problem of interpretable dialogue modeling. The hierarchical behavior taxonomy is grounded in established linguistic theory.

**Weaknesses:**

**Mismatch Between Claims and Implementation**

This is correlation mining, not causal inference in the formal sense. The paper repeatedly claims to perform "causal inference" but the implementation uses frequency-based co-occurrence graphs.The adjacency matrix is symmetric (undirected graph), but causation has inherent directionality (A causes B does not imply B causes A). Without directed edges, the graph cannot represent causal pathways. Furthermore, observational correlation does not distinguish causation from confounding. Required evidence for causal claims would include: intervention experiments (removing graph nodes and measuring outcome changes), counterfactual generation (reasoning about what would have happened under different conditions), or Granger causality tests (demonstrating that history of X improves prediction of Y beyond Y's own history). The paper contains none of these analyses.

**The system architecture separates prediction from explanation.**

The pipeline first predicts behavior labels, then generates rationales conditioned on those predictions. This is post-hoc explanation generation rather than reasoning that drives decision-making.

**High-level and low-level behaviors are trained independently despite clear pragmatic dependencies.**

The training loss and inference approximation assume conditional independence. However, pragmatic constraints create strong dependencies. For instance, if high-level intent is Acknowledgment, low-level action is unlikely to be Interruption—these are pragmatically inconsistent. Similarly, Directive intents typically pair with specific low-level actions like Turn-taking or Continuation.Conversation Analysis literature documents structured patterns like adjacency pairs (Question → Answer) and preference organization (Offers preferentially receive Acceptances; Rejections are dispreferred and marked by delays and hedges). The independent modeling cannot capture these dependencies.

**The synthetic corpus exhibits systematic biases that may limit generalization.**

Statistical analysis reveals significant differences from human baselines.

**Missing phenomena** include repair sequences (self-correction, clarification requests), emotion dynamics (frustration escalation, sarcasm), dispreferred actions (rejections requiring hedges and accounts), and social context (power dynamics, cultural variation). The TTS synthesis cannot capture authentic prosodic variation in emotion or stance.

**Multiple components use fixed rules that may not scale to new domains or contexts.**

Low-level label assignment uses priority rules: Backchannel > Interruption > Turn-taking > Continuation. These priorities are hand-coded rather than learned from data and may not reflect actual pragmatic hierarchies across different contexts. Graph edges are constructed purely from co-occurrence counts without considering semantic relation strength, temporal decay (recent events should weigh more), or learned edge probabilities. GAT depth is fixed, limiting reasoning to bounded graph traversal. The system cannot adapt to domain-specific speech acts.

**Human Evaluation Details**

Critical methodological details are missing from the human evaluation protocol. Without blinding, raters knowing which system generated which audio introduces confirmation bias. Without randomization, order effects may occur. Without inter-rater reliability measures, score reliability cannot be assessed.

**Unreported information**: Sample size, number of raters, inter-rater agreement statistics, rater expertise level, and evaluation procedure.

**Questions:**

**Q1**: Given that your graph construction uses symmetric co-occurrence matrices without intervention operators or counterfactual mechanisms, can you clarify what "causal inference" means in your framework?

**Q2**: The system predicts behaviors then generates explanations. How do you distinguish this from post-hoc rationalization? Have you tested whether explanations remain valid when features they mention are removed (faithfulness tests)?

**Q3**: Why are high and low levels trained independently when pragmatic constraints create dependencies? Have you measured how often predictions are pragmatically inconsistent (e.g., Acknowledgment + Interruption)?

**Q4**: What is the measured end-to-end latency? (Please provide more data to explain.)

**Q5**: Have you tested on adversarial cases like sarcasm, rhetorical questions, emotional speech, or multi-party conversations? What are concrete failure examples?

**Q6**: Why use symmetric undirected graphs when causation is directional? Have you compared undirected co-occurrence versus directed learned edges?

---

### Official Review · Reviewer_dbRG · 2025-11-01

**Soundness:** 2
**Presentation:** 1
**Contribution:** 2
**Rating:** 2
**Confidence:** 2

**Summary:**

This paper introduces a conversational behavior reasoning framework for full-duplex speech systems that explicitly models the causal relations between high-level communicative intents (e.g., constatives, directives, acknowledgments) and low-level speech acts (e.g., turn-taking, backchannel, interruption). The proposed Graph-of-Thought (GoT) architecture formalizes the perception–reasoning–generation loop in conversation, enabling interpretable predictions and rationale generation. Experiments on both synthetic duplex dialogues and real conversational speech (Candor corpus) demonstrate that the framework achieves strong performance on hierarchical speech-act detection and generates coherent, causal rationales for its decisions.

**Strengths:**

The paper proposes a meaningful shift from black-box sequence prediction to causal reasoning over conversational behavior, arguing that next-behavior reasoning is a more human-aligned formulation for full-duplex systems.

The paper carefully constructs a dataset combining a simulation corpus with real data.

**Weaknesses:**

1. Methodology

The method section is difficult to follow in parts. It is unclear how OpenIE triples (subject–relation–object) are incorporated into the graph and how they interact with the speech-act nodes.

The paper describes multiple node types (text nodes, high-level acts, low-level acts) but does not provide a clear illustrative example showing how these are connected or how causal dependencies are inferred.

The rationale generation mechanism could be better illustrated with an explicit example mapping audio input → nodes → rationale text.

2. Data and Baselines

The distribution and composition of the real dataset (Candor corpus) should be described in more detail (e.g., number of speakers, domains, or event ratios).

The paper lacks baseline comparisons for both behavior detection and reasoning tasks.

The training setup could be clarified: while encoders (HuBERT, Whisper, T5, GAT) are mentioned, it is not clear which components are frozen or what models are fine-tuned, and what parameter scales are used for each experiment.

3. Experimental Results

In Figure 3, the “Audio-only” modality appears to outperform “Audio + Text” or “Audio + Text + Graph” configurations, which contradicts the expected benefit of multimodal reasoning.

The metrics reported in Section 6.3 and Table 6 (human ratings) seem inconsistent — e.g., the mean ratings differ slightly between the text and table — and should be double-checked.

The evaluation focuses on BLEU/ROUGE and semantic similarity, but lacks human qualitative analysis or ablations on causal reasoning quality.

**Questions:**

NA

---

### Note · Authors · 2026-01-24

I have read and agree with the venue's withdrawal policy on behalf of myself and my co-authors.